# A florigen paralog is required for short-day vernalization in a pooid grass

**Daniel Woods[1,2,3†]\*, Yinxin Dong[3,4‡], Frederic Bouche[3§], Ryland Bednarek[3#], Mark Rowe[3], Thomas Ream[2,3¶], Richard Amasino[1,2,3]\***

[1]Laboratory of Genetics, University of Wisconsin, Madison, United States; [2]United States Department of Energy Great Lakes Bioenergy Research Center, University of Wisconsin-Madison, Madison, United states; [3]Department of Biochemistry, University of Wisconsin, Madison, United states; [4]College of Horticulture, Northwest A&F University, Yangling, China

**Abstract** Perception of seasonal cues is critical for reproductive success in many plants. Exposure to winter cold is a cue that can confer competence to flower in the spring via a process known as vernalization. In certain grasses, exposure to short days is another winter cue that can lead to a vernalized state. In *Brachypodium distachyon*, we find that natural variation for the ability of short days to confer competence to flower is due to allelic variation of the *FLOWERING LOCUS T* (*FT1*) paralog *FT-like9* (*FTL9*). An active *FTL9* allele is required for the acquisition of floral competence, demonstrating a novel role for a member of the *FT* family of genes. Loss of the short-day vernalization response appears to have arisen once in *B. distachyon* and spread through diverse lineages indicating that this loss has adaptive value, perhaps by delaying spring flowering until the danger of cold damage to flowers has subsided.
DOI: https://doi.org/10.7554/eLife.42153.001

**\*For correspondence:**
dpwoods@wisc.edu (DW);
amasino@biochem.wisc.edu (RA)

**Present address:** [†]Plant Sciences Department, University of California-Davis, Davis, United States; [‡]Key Laboratory for Plant Diversity and Biogeography of East Asia, Kunming Institute of Botany, Chinese Academy of Sciences, Kunming, China; [§]Laboratory of Plant Physiology, University of Liege, Liege, Belgium; [#]Boyce Thompson Institute, Cornell University, Ithaca, United States; [¶]Monsanto Company, Chesterfield, United States

## Introduction

Many plants adapted to temperate climates have a biennial or winter-annual life history strategy. These plants become established in the fall, overwinter, and flower in the spring. Essential to this adaptive strategy is that flowering does not occur prior to winter, and that the perception of winter leads to competence to flower in the spring (*Woods et al., 2014b*). The winter cue perceived in many plants is exposure to a prolonged period of cold, and the process by which such exposure leads to competence to flower is known as vernalization (*Chouard, 1960*; *Woods et al., 2014a*). In some plants, however, short days (SD) provide an alternative winter cue (*Purvis and Gregory, 1937*; *Heide, 1994*), and the process by which SD exposure leads to competence to flower has been referred to as SD vernalization (*Purvis and Gregory, 1937*). This phenomenon was called SD vernalization because a hallmark of cold-mediated vernalization is acquisition of *competence* to flower rather than flowering per se, and the SD-vernalization phenomenon is similar in that exposure to SD leads to competence, but plants must still be exposed to inductive LD to flower.

The physiology of SD vernalization has been studied extensively in rye, wheat, barley, and oat (*Purvis and Gregory, 1937*; *Dubcovsky et al., 2006*; *Sampson and Burrows, 1972*; *Heide, 1994*) although it exists in other families of plants as well (e.g., *Chouard, 1960*). The study of SD vernalization in wheat and barley is complicated by the fact that there are genotypes of wheat (e.g. Templar) and barley (e.g. Morex) in which flowering per se occurs in SD; that is certain wheat and barley genotypes are facultative LD plants in that they flower most rapidly in LD but also will flower in SD (*Kikuchi et al., 2009*; *Casao et al., 2011a*; *Evans, 1987*). This SD flowering in wheat and barley is distinct from SD vernalization. *Brachypodium distachyon* (*B. distachyon*), however, is an obligate LD plant that only has a SD-vernalization pathway and not a SD-flowering pathway (*Ream et al., 2014*;

*Gordon et al., 2017*; *Woods and Amasino, 2015*); thus, the SD-vernalization pathway can be studied in *B. distachyon* without the complication of SD flowering.

Little is known at a molecular level about how the SD-vernalization pathway operates in any plant species, and thus we have explored the genetic basis of SD vernalization in *B. distachyon*. Our work reveals a novel role for a *FT*-like gene in the SD-vernalization pathway and provides a molecular explanation of the distinction between SD vernalization and SD flowering in the pooid grasses.

## Results and discussion

### Natural variation in the SD vernalization response in *B. distachyon*

The model pooid grass B. distachyon has a robust cold-mediated vernalization response (*Ream et al., 2014*; *Gordon et al., 2017*). To determine if any accessions also have a SD vernalization response, we grew 51 accessions in 8 hr SD followed by long days (LD; eight accessions shifted into 14 hr LD *Figure 1—figure supplements 1A* 43 accessions into 16-h LD (*Supplementary file 1*) or 20 hr (*Supplementary file 2*)); as controls accessions were grown solely in LD or SD. Flowering time was measured as days to heading and leaf number on the parent culm. Furthermore, dissections on the parent culm were made at the end of the experiment in accessions that did not visibly flower to determine the developmental state of the meristem. Forty of the accessions exhibited a robust SD-vernalization response: these accessions flowered rapidly after the shift from SD to LD, but in LD alone flowering occurred only after quite long periods of growth. The accessions never flowered when grown solely in SD (*Figure 1A and B*, *Supplementary file 1 and S2*, *Figure 1—figure supplement 1*; Materials and methods contains the rationale for the controls); furthermore, dissections on the parent culm revealed that the meristems were still vegetative after 150 days of growth in SD in all 51 accessions tested, consistent with our previous studies (*Ream et al., 2014*; *Gordon et al., 2017*) indicating that B. distachyon is an obligate LD plant. We refer to these 40 lines as SD-vernalization responsive and the 11 lines that did not have a SD vernalization response as SD-vernalization non-responsive.

Given that the longest days experienced in the native growth habitats of the *B. distachyon* accessions tested range between 15 and 16 hr (*Figure 1—figure supplement 2*), we evaluated whether the SD vernalization response was still effective when SD-grown accessions Koz3, TR12c and RON2 were shifted into a range of more native photoperiods (*Figure 1D*). Plants were grown in 8 hr SD for 9 weeks before shifting into 10, 12, 14, 15, or 16 hr day-lengths (*Figure 1D*). All three SD-responsive accessions flowered more rapidly after exposure to SD when the day-length was 12 hr or longer (*Figure 1D*). In *B. distachyon* 12 hr is the minimal day-length that is still partially inductive for flowering (*Ream et al., 2014*; *Woods et al., 2017a*). Thus, the flowering effect of SD vernalization is manifest at the same inductive photoperiods as cold-mediated vernalization.

To determine if SD vernalization, like cold-mediated vernalization, is a quantitative response, we conducted a SD-exposure time course. Four SD-responsive and five non-responsive accessions were grown for 2, 4, 6, 8, 10, or 12 weeks in 8 hr SD before shifting into LD. Like cold-mediated vernalization, the SD response is quantitative—longer periods of SD exposure result in more rapid flowering in the SD-responsive accessions (*Figure 1E*; *Figure 1—figure supplement 1C*). Furthermore, 8 weeks of SD exposure saturates the SD-vernalization response. Lastly, even 12 weeks of SD exposure did not enable flowering in any of the accessions we had characterized as SD non-responsive in the 8 week SD exposure evaluation noted above.

We also explored whether the SD vernalization response is possible at a range of developmental stages. In the experiments described above, the SD treatment started when imbibed seeds were sown in soil and continued until plants were shifted into LD and, thus, depending upon the length of time in SD, the SD treatment spanned a range of early developmental stages. To determine if SD vernalization is effective at later developmental stages, we grew SD-responsive accessions in LD for 2, 4, and 6 weeks and subsequently exposed them to 8 weeks of SD. Plants were then returned to inductive LD to determine their SD vernalization responsiveness. All of the SD vernalized plants flowered within 35 days after shifting back into LD, with four additional leaves appearing on the parent culm before flowers were visible (*Figure 1F*; *Figure 1—figure supplement 1D,E*), indicating the SD vernalization is equally effective at early and later developmental stages.

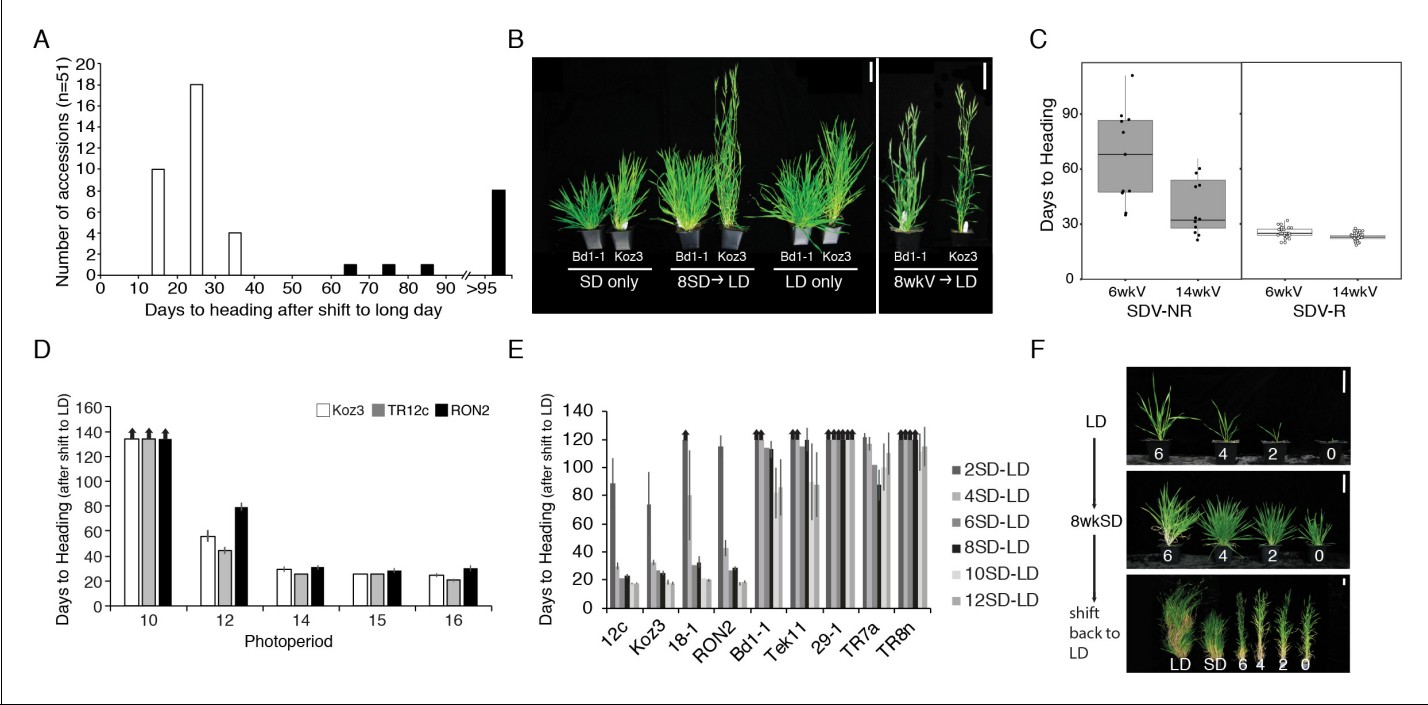

**Figure 1.** Natural variation in the short-day (SD) vernalization response in *B.distachyon*. (**A**) SD vernalization response in 51 accessions that require cold mediated vernalization. Plants were grown for 56 days in 8 hr SD before the shift into long days (LD) of 14 hr (eight rapid flowering accessions when grown in 16 hr or 20 hr, see also *Figure 1—figure supplement 1*) or 16 hr (43 delayed flowering accessions). Day temperature of 21°C and night temperature of 18°C. SD vernalization is equally effective when day and night temperatures are constant (see Materials and methods for details). White bars indicate accessions that are SD vernalization responsive and black bars accessions that are SD vernalization non-responsive. (**B**) Image of a representative SD vernalization responsive accession Koz3 and a SD vernalization non-responsive accession Bd1-1 taken after 100 days of growth. Only Koz3 flowers after the SD to LD shift whereas both accessions flower rapidly if exposed to 8 weeks of cold (8wkV) prior to growth in 16 hr LD. (**C**) Box plot illustrating that SD vernalization non-responsive accessions (SDV-NR, black dots) require longer periods of cold exposure to flower rapidly in 16 hr LD relative to SD vernalization responsive (SDV-R, white dots) accessions. The difference between SDV-NR and SDV-R flowering after vernalization is statistically significant. See supplemental Table S1-2 for days to heading, leaf count and standard deviation data for SD vernalization and Table S3 for days to heading data for 6 and 14 weeks of vernalization (6wkV and 14wkV). Bar = 5 cm (**D**) Short day vernalization response in three delayed-flowering accessions Koz3, TR12c and RON2 when shifted into 10, 12, 14, 15, and 16 h days. In this experiment, LD only and SD only control plants flowered similar to those in Table S1-2 (data not shown). Arrows indicate treatments in which plants did not flower within the experiment. Bars represent the average days to heading of 12 plants for each treatment. (**E**) SD vernalization time course. See *Figure 1—figure supplement 1B* for SD and LD only controls and *Figure 1—figure supplement 1C* for representative photo of plants at the end of the experiment. (**F**) The SD vernalization response is effective at multiple developmental stages. Koz3 and Tr12c were grown for 6, 4, 2, 0 weeks under 16 hr LD before shifting into 8 weeks of 8 hr SD (8wkSD). After the SD treatment plants were shifted back into LD. Representative photo of Koz3 taken after 40 days. See *Figure 1—figure supplement 1D,E* and Supplemental Table 1 and 2 for days to heading and leaf count data.

DOI: https://doi.org/10.7554/eLife.42153.002

The following figure supplements are available for figure 1:

**Figure supplement 1.** Natural variation in the short day vernalization response.
DOI: https://doi.org/10.7554/eLife.42153.003

**Figure supplement 2.** Geographical distribution of *B.distachyon* accessions used in this study.
DOI: https://doi.org/10.7554/eLife.42153.004

## Genetic mapping of the SD vernalization response identifies an FT paralog

To explore the genetic basis of natural variation in the SD-vernalization response, we generated mapping populations from crosses between responsive and non-responsive accessions (Koz3 X Bd1-1; 12 c X Bd1-1; RON2 X Bd1-1; and Koz3 × 29–1; *Figure 2—figure supplement 1*). In all populations, SD responsiveness segregated as a single, dominant locus (*Figure 2—figure supplement 1A*) that mapped as a large-effect QTL near the bottom of chromosome 2 (*Figure 2—figure supplement 1B*; *Figure 2—source data 1*). Fine mapping using the Koz3 X Bd1-1 F2 population narrowed

the interval to a region on chromosome two containing eight annotated genes (*Figure 2—figure supplement 1C*). None of these genes exhibited a difference in expression pattern in the shoot apical meristem or leaf tissue of responsive versus non-responsive lines grown in SD followed by LD, SD only, or LD only (data not shown), indicating that the variant underlying this trait is not likely to be in a cis-regulatory region of a gene. However, there is one gene in the interval that is preferentially expressed in SD, and this gene exhibits a one nucleotide change in all 11 SD non-responsive accessions that is not found in any of the 40 SD-responsive accessions; this change results in a threonine to lysine substitution at position 94 (T94K) in an *FT* paralog (*Supplementary file 1–2*; *Figure 2—figure supplement 2*; *Figure 2—source data 2*) referred to as *FT-like 9* (Bradi2g49795; *FTL9*) (*Higgins et al., 2010*; *Figure 2—figure supplement 4*). The founding member of this family is *FLOWERING LOCUS T* (*FT*) which encodes a small protein with similarity to phosphatidylethanolamine-binding proteins, also known as 'florigen', that travels from leaves to the shoot apical meristem to induce flowering (e.g., *Corbesier et al., 2007*; *Tamaki et al., 2007*).

Molecular analyses confirm that allelic variation at *FTL9* is responsible for natural variation in the ability of SD exposure to confer competence to flower (*Figure 2*; *Figure 2—figure supplement 1*). First, reducing *FTL9* expression in the SD-responsive accessions Koz3 and Bd21-3 using artificial microRNAs (amiRNAs) eliminates the SD-vernalization response (*Figure 2A* and *Figure 2—figure supplement 1D–I*). However, the amiRNA lines remain fully responsive to cold-mediated vernalization demonstrating that the role of *FTL9* is specific to the SD-vernalization pathway (*Figure 2B* and *Figure 2—figure supplement 1E–F*). Second, constitutive expression of an *FTL9* cDNA from the SD-responsive accessions Koz3 and Bd21 under the control of the maize ubiquitin promoter in both SD-responsive (Bd21-3 and Koz3) and non-responsive accessions (Bd29-1), which results in *FTL9* expression in LD, causes LD-grown transgenic plants to behave as if they were vernalized without prior cold or SD treatment both in terms of accelerated flowering as well as expression of *VERNALIZATION1* (*VRN1*), a MADS box transcription factor related to the *APETALA/FRUITFUL* family of genes in *Arabidopsis thaliana*, that is a marker of the vernalized state in grasses (*Woods et al., 2014a*; *Yan et al., 2003*; *Preston and Kellogg, 2008*) (*Figure 2C* and *Figure 2—figure supplement 3*). In contrast, constitutive expression of the *FTL9* cDNA from the SD-non-responsive accession Bd1-1 did not affect the flowering behavior of transgenic lines (*Figure 2C*; *Figure 2—figure supplement 3*) indicating that the T94K change is likely to disrupt *FTL9* function consistent with the recessive nature of this allele (*Figure 2—figure supplement 1A*) and that the T94K change occurs in a highly conserved amino acid in this family of proteins in plants and animals (*Figure 2—figure supplement 2B*). In the experiments described above we evaluated transgenic expression of *FTL9* alleles from both Koz3 and Bd21-3 because *FTL9* alleles from SD-responsive accessions exhibit variation in the predicted C terminus of the FTL9 protein (*Figure 2—source data 2*): the ancestral state of FTL9 is 178 amino acids, but in Koz3 there is a deletion of a single nucleotide (G) in amino acid 175 that alters the reading frame such that FTL9 is 184aa long. As noted above, both versions of *FTL9* confer competence to flower.

Overexpression of *FTL9* does not result in rapid flowering when plants are grown in SD whereas overexpression of *FT1* results in rapid flowering in SD (*Figure 2—figure supplement 3*). The lack of rapid flowering in the *FTL9* overexpression transgenics is consistent with the gene providing competence to flower rather than flowering per se, whereas *FT1* expression directly induces flowering in all growth conditions (*Figure 2—figure supplement 3*; *Woods et al., 2017a*).

## Loss of the SD-vernalization response arose once in B. distachyon

Although all SD non-responsive accessions contain the T94K change at the *FTL9* locus, they do not cluster in a single group but rather are quite distantly related based on whole-genome analyses (*Gordon et al., 2017*; *Figure 2—figure supplement 5A*) raising the possibility that the T94K allele arose independently more than once. However, a phylogenetic analysis focusing on the 65 kb interval containing *FTL9* indicates that in all of the SD-non-responsive accessions the 65 kb *FTL9* interval is highly conserved (*Figure 2—figure supplement 5B*) despite other regions of the genome being divergent. This suggests that the *FTL9* variant associated with loss of SD responsiveness arose once and was then maintained by positive selection throughout outcrossing to diverse accessions. Although, *B. distachyon* is typically inbreeding, outcrossing occurs as well (*Sancho et al., 2018*). That a lesion in *FTL9* that appears to have undergone positive selection arose once is in contrast to the loss of a vernalization requirement in *Arabidopsis thaliana* which arose independently many

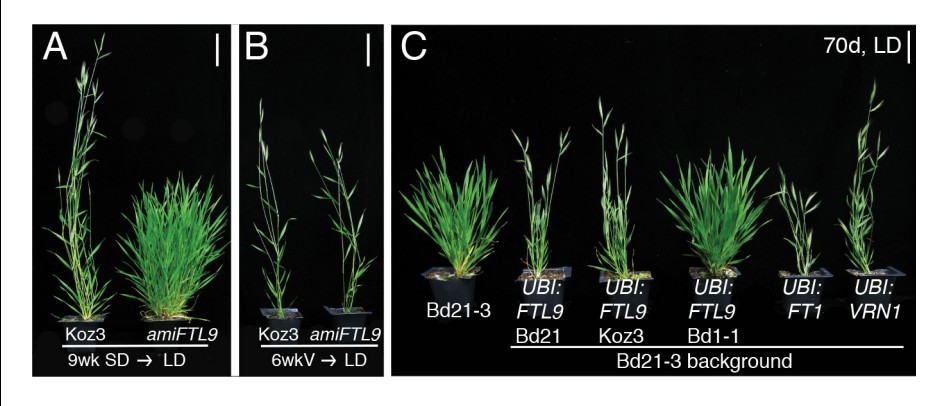

**Figure 2.** *FTL9* is necessary for the SD vernalization response and sufficient to confer competence to flower. (**A**) *amiFTL9* knockdown prevents SD vernalization. Koz3 wild-type and *amiFTL9* knockdown plants grown in 8 hr SD for 9 weeks (9wk SD) before shifting into 20 hr LD. Bar= 5 cm. (**B**) *amiFTL9* knockdown has no effect on cold-mediated vernalization. Plants were vernalized as imbibed seed at 5°C for 6 weeks (6wkV) before outgrowth in 20 hr LD. Bar = 5 cm. See *Figure 2—figure supplement 1* for *amiFTL9* details. All experiments were repeated with similar results. (**C**) Constitutive expression of dominant FTL9 alleles permits flowering in LD. Representative photo of Bd21-3 wild-type, *UBI:FTL9* Bd21 (*FTL9* from Bd21), *UBI:FTL9* Koz3 (FTL9 from Koz3), *UBI:FTL9* Bd1-1 (FTL9 from Bd1-1), *UBI:FT1*, and *UBI:VRN1* grown in a 16 hr photoperiod (LD) without cold or short day vernalization. All over expression lines are in the Bd21-3 background. Bar = 5 cm. See *Figure 2—figure supplement 3* for *UBI:FTL9* details including days to heading, leaf count data as well as mRNA expression analyses.

DOI: https://doi.org/10.7554/eLife.42153.005

The following source data and figure supplements are available for figure 2:

**Source data 1.** Koz3 X Bd1-1 F2 QTL information.
DOI: https://doi.org/10.7554/eLife.42153.012
**Source data 2.** VCF summary of variants in mapped interval.
DOI: https://doi.org/10.7554/eLife.42153.013
**Figure supplement 1.** Identification of the *FT*-like paralog *FTL9* as essential for SD vernalization.
DOI: https://doi.org/10.7554/eLife.42153.006
**Figure supplement 2.** Coding variant summary within the SD vernalization mapped interval between SD vernalization responsive and non-responsive accessions.
DOI: https://doi.org/10.7554/eLife.42153.007
**Figure supplement 3.** Effects of ubiquitin promoter-mediated constitutive expression of *FTL9*, *FTL10*, *FT1*, and *VRN1*.
DOI: https://doi.org/10.7554/eLife.42153.008
**Figure supplement 4.** Maximum likelihood phylogenetic relationships among a subset of *FT*-like genes based on a nucleotide alignment of the PEBP domain.
DOI: https://doi.org/10.7554/eLife.42153.009
**Figure supplement 5.** Population analysis.
DOI: https://doi.org/10.7554/eLife.42153.010
**Figure supplement 6.** Analysis of syntenic chromosomal regions containing *FTL9* orthologs in grasses.
DOI: https://doi.org/10.7554/eLife.42153.011

times due to loss-of-function mutations in *FRIGIDA* (*Le Corre et al., 2002*). Perhaps the T94K change does not result in a complete loss of function, but rather retains a function distinct from its role in SD vernalization.

The adaptive value of an active *FTL9* and a SD-vernalization response, which is the ancestral state, may be to ensure that vernalization occurs in mild climates in which SD may be a more reliable indicator of winter than cold. The adaptive value of loss of *FTL9* activity and the corresponding loss of the SD-vernalization response might be to enable *B. distachyon* to grow in regions with a more variable spring climate in which a robust SD-vernalization response might lead to flowering before the danger of a hard freeze, which would damage sensitive floral organs, had passed. Interestingly, the SD-non-responsive accessions also require a longer period of cold exposure than the SD-responsive

accessions to become fully vernalized (*Gordon et al., 2017* and *Figure 1C*; *Supplementary file 3*) which is consistent with the SD-non-responsive accessions having undergone adaptation for later spring flowering. Also, consistent with this model of the adaptive value of loss of *FTL9* activity, the non-responsive accessions tend to have been collected at higher latitudes than the responsive accessions (*Figure 1—figure supplement 2*); however, latitude is only one of many factors that may relate to the latest date of a damaging freeze.

## Characterization of FTL9 expression

Consistent with its role in SD vernalization, *FTL9* mRNA is barely detectable in LD, whereas its expression is 350-fold greater in SD (*Figure 3A*). When SD-grown plants are shifted to LD, *FTL9* mRNA levels decline within a few days (*Figure 3A*). We tested whether the quantitative aspect of SD vernalization might be correlated with increasing expression of *FTL9* during the course of growth in SD, but *FTL9* mRNA levels remained the same over 10 weeks of SD exposure (*Figure 3—figure supplement 1D*). Finally, *FTL9* exhibits a diurnal expression pattern in SD with mRNA levels increasing during the night to a peak at dawn and declining during the day to a minimum at dusk. In LD there is no detectable diurnal expression difference; *FTL9* expression remains low throughout a 20 hr LD (*Figure 3B* and *Figure 3—figure supplement 1A*). When plants are shifted from SD to free-run, constant-dark conditions, *FTL9* mRNA levels plummeted and did not oscillate, indicating that diurnally oscillating *FTL9* expression requires short day/long night cycles (*Figure 3—figure supplement 1B*).

## Roles of SD expressed FT-like genes in grass flowering

*FTL9* is part of a *FT*-like gene family that has expanded to 14 members in grasses (*Higgins et al., 2010* and *Figure 2—figure supplement 4*). The roles of the other *FT*-like genes in grasses have not been thoroughly explored. Studies in wheat and barley indicate that a candidate gene underlying a QTL that confers the ability of certain varieties to flower in SD (*Ppd-H2*) (*Laurie et al., 1995*) is a paralog of *FT* referred to as *FT3* (*Faure et al., 2007*; *Kikuchi et al., 2009*; *Casao et al., 2011a*). The dominant *FT3* allele, which enables SD flowering, is often found in spring barley cultivars grown in more southern latitudes, whereas the recessive allele is typically associated with winter cultivars grown in more northern latitudes (*Casao et al., 2011a*). *FT3* is implicated in actual flowering in SD in barley as opposed to the competence to flower in LD that is conferred by SD-specific expression of *FTL9* in *B. distachyon*. *FTL10* (Bradi2g19670) is the *B. distachyon* ortholog of barley *FT3* (*Higgins et al., 2010*; *Halliwell et al., 2016*; *Figure 2—figure supplement 4*). *FTL10* and *FTL9* are closely related paralogs that resulted from a grass-specific duplication event, and thus *FTL10* and *FTL9* reside in sister clades (*Figure 2—figure supplement 4*). *FT3* is up-regulated in SD in barley (*Faure et al., 2007*) and wheat (*Lv et al., 2014*), whereas *FTL10* expression in *B. distachyon* is not modulated by day-length and its expression level is quite low in all photoperiod and temperature conditions tested (*Figure 3D*). Due to its minimal expression, *FTL10* may not play a role in *B. distachyon* flowering. However, constitutive expression of *FTL10* from the maize ubiquitin promoter results in flowering during transgenic line generation in tissue culture (*Figure 3E*) and rapid flowering in SD (*Figure 2—figure supplement 3I,J*) similar to the effects of expression of other *FT1* orthologs in *B. distachyon* (*Figure 2—figure supplement 3I,J*; *Ream et al., 2014*) and wheat (*Lv et al., 2014*) indicating that *FTL10* has the ability to induce flowering. Thus, the *FT3/FTL10* orthologs in this grass-specific duplication event have diverged in barley and wheat versus *B. distachyon*. In barley and wheat, *FT3* has retained its presumably ancestral role of being expressed in SD which leads to flowering in SD. *B. distachyon* cannot flower in SD because its ortholog, *FTL10*, is no longer expressed. As discussed above, constitutive expression in *B. distachyon* of the other duplicated gene, *FTL9*, does not induce flowering in SD or in tissue culture because its role is to establish competence to flower as opposed to the florigen-like role of *FTL10/FT3* in inducing flowering (*Figure 2—figure supplement 3G,I,J*).

SD-expressed *FT* family members emerged early in grass diversification. For example, like *FTL9* in *B. distachyon*, the *FTL9* ortholog in sorghum and maize (*CENTRORADIALIS 12*; *CN12*) (*Figure 2—figure supplement 4*, *Figure 2—figure supplement 6*) is also expressed only in SD (*Murphy et al., 2011*; *Meng et al., 2011*; *Wolabu et al., 2016*). However, in sorghum and maize, which are SD-flowering or day neutral plants, *CN12* and SD expressed *CN8* (the *FTL10/FT3* ortholog) are both

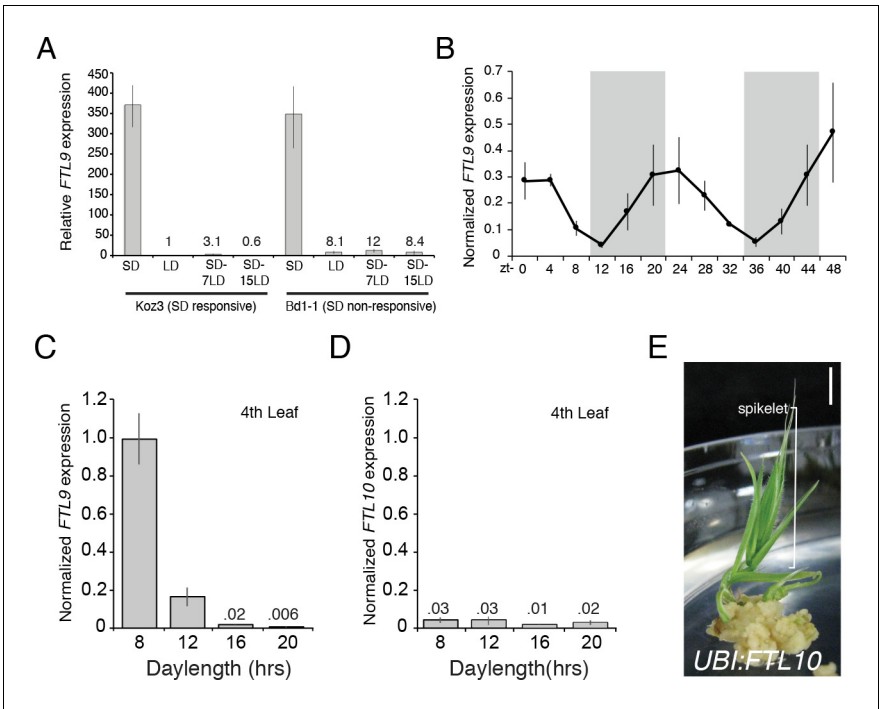

**Figure 3.** *FTL9* expression is SD specific and diurnally fluctuating whereas *FTL10* expression is not influenced by day-length and encodes a florigenic protein. (A) *FTL9* mRNA levels in Koz3 and Bd1-1 grown solely in 8 hr SD, solely in 20 hr LD, or 7 or 15 days after a shift from 8 weeks in SD to LD. Dissection of meristems on the parent culm revealed that all remained meristems vegetative in both Koz3 (12 dissected meristems) and Bd1-1 (12 meristems) in SD and LD grown plants. After a shift from SD to LD at day 7 all of the meristems were vegetative in Koz3 and Bd1-1 (12 meristems for each accession), however by day 15 in LD all of the meristems had converted to floral meristems in Koz3 (12 meristems) whereas they remained vegetative in Bd1-1 (12 meristems). RNA was prepared from newly expanded leaves on the parent culm. (B) Diurnal *FTL9* mRNA fluctuations in 8 hr SD. Plants were grown in SD until the fourth-leaf stage was reached at which point newly expanded leaves were harvested every 4 hr throughout a 48 hr diurnal cycle. Shaded boxes represent dark periods. (C) *FTL9* and (D) *FTL10* mRNA levels in 8, 12, 16, and 20 hr photoperiods. Koz3 was grown to the four-leaf stage and newly expanded leaves were harvested in the middle of the photoperiod. (E) Representative image of T0 generation *UBI:FTL10* plants showing rapid spikelet formation (bracket) on callus regeneration media. All 15 independent *UBI:FTL10* calli flowered rapidly in regeneration media. Those plants without the transgene did not flower rapidly. *UBI:FTL10* T1 generation days to heading and leaf count data see *Figure 2—figure supplement 3*. Bar = 0.5 cm. (A–D) Values represent the average of three biological replicates ± standard deviation (four leaves per replicate). Similar results were obtained in independent experiments. Gene expression was normalized to *UBC18* as described in *Ream et al. (2014)*.

DOI: https://doi.org/10.7554/eLife.42153.014

The following figure supplement is available for figure 3:

**Figure supplement 1.** *FTL9* and *FTL10* diurnal mRNA expression.

DOI: https://doi.org/10.7554/eLife.42153.015

florigenic similar to *FTL10/FT3* (*Meng et al., 2011*; *Lazakis et al., 2011*; *Wolabu et al., 2016*). Thus, gene duplication of an ancestral SD-expressed *FT*-like gene at the base of the grass family appears to have resulted in the *FTL9/CN12* and *FTL10/FT3/CN8* clades (*Figure 2—figure supplement 4*). Grasses that are classified as SD plants like sorghum and photoperiodic maize (*Murphy et al., 2011*; *Meng et al., 2011*) are in fact SD plants because a florigenic member of one of the SD-expressed clades provides the primary florigen activity. Wheat and barley are LD plants because the strongest florigen activity is provided by LD-expressed *FT* family members in other clades such as *FT1*. However, many varieties of wheat and barley exhibit a facultative LD response—that is they also can flower in SD. The SD flowering of barley appears to result from expression of SD-induced clade members such as *FT3* (*Kikuchi et al., 2009*; *Faure et al., 2007*). There are other barley varieties that

do not flower in SD because their *FT3* alleles have lesions in the coding region (*Kikuchi et al., 2009*). All accessions of *B. distachyon* that we have analyzed are obligate LD plants because the florigenic member of the clade, *FTL10*, is not expressed at sufficient levels to cause flowering in SD and the other member of the clade, *FTL9*, evolved the different function of providing competence to flower rather than directly promoting flowering like it does in sorghum and maize.

SD-mediated vernalization was first described in rye in 1937 (*Purvis and Gregory, 1937*) and has been described in wheat and barley (*Evans, 1987*; *Heide, 1994*; *Dubcovsky et al., 2006*); however, sequence data available to date have not revealed a clear ortholog of *FTL9* in wheat or barley (*Figure 2—figure supplement 4*; *Faure et al., 2007*). It will be interesting to determine whether *FTL9* orthologs or other *FT* family members are involved in SD vernalization in cereals.

Gene duplication and divergence has enabled diverse roles for *FT* family members in flowering. The founding members of the *FT* family provide florigen activity (*Corbesier et al., 2007*; *Tamaki et al., 2007*). In beet, an *FT* family member is an inhibitor of flowering that is key to establishing a requirement for vernalization (*Pin et al., 2010*). Our work in *B. distachyon* reveals a new role for an *FT* family member: the establishment of competence to flower without direct florigen activity.

## FTL9 is repressed in LD by VRN2

Given the day-length dependence of *FTL9* expression, we analyzed its expression in a *phyC-1* mutant background (*Woods et al., 2014a*). PHYTOCHROME C (PHYC) is a red light receptor and in *phyC* mutants the LD photoperiod flowering pathway is abolished and mutant plants are extremely delayed in flowering and have the appearance of SD-grown plants when grown under LD (*Woods et al., 2014a*). Indeed, *FTL9* expression is significantly elevated in *phyC* mutants grown in LD compared to wild type across all time points tested (*Figure 4A*). Thus, certain PHYC-regulated genes may be negative upstream regulators of *FTL9*. One candidate gene is *FT1*; in barley cultivars that can flower in SD, *HvFT1* has been postulated to be a repressor of *HvFT3*—the SD-expressed paralog of *FTL9* discussed above (*Figure 2—figure supplement 4*) —because relatively high *HvFT3* expression in SD precedes the expression of *HvFT1* and, as *HvFT1* expression levels increase in SD, *HvFT3* expression decreases (*Kikuchi et al., 2009*). The same correlation of expression patterns exists in *B. distachyon*; for example, in the *phyC* mutant in LD, *FT1* is nearly undetectable (*Woods et al., 2014a*), and *FTL9*, as noted above, is highly expressed, whereas in wild type *FT1* is expressed in LD and *FTL9* is not. To determine if *FT1* represses *FTL9* in *B. distachyon*, we evaluated *FTL9* expression in *UBI:FT1* transgenic lines grown in SD, a condition in which *BdFT1* is not expressed in wild-type plants. *FTL9* expression levels were similar between *UBI:FT1* and wild-type (*Figure 2—figure supplement 3H*); thus, it is unlikely that *FT1* acts as a LD repressor of *FTL9* in *B. distachyon*.

*VERNALIZATION2* (*VRN2*) which contains a putative zinc finger and a *CONSTANS, CONSTANS*-like, and *TIMING OF CAB1* domain (*Yan et al., 2004*) is another candidate gene that might repress *FTL9* during LD because *VRN2* expression is abolished in the *phyC* mutant (*Woods et al., 2014a*) and the level of *FTL9* expression is inversely correlated with the level of expression of *VRN2* (*Figure 4B* and *Figure 4—figure supplement 1A*). *VRN2* is a flowering repressor, and its expression is a function of day-length (*Figure 4B* and *Figure 4—figure supplement 1A*) (*Dubcovsky et al., 2006*; *Woods et al., 2016*). In LD, *VRN2* mRNA levels are elevated and *FTL9* mRNA is barely detectable, whereas in SD *VRN2* mRNA levels are low and *FTL9* is highly expressed (*Figure 4—figure supplement 1A*). When plants are shifted from SD to LD, *VRN2* levels rise to LD levels (*Figure 4B*), and *FTL9* expression subsides (*Figure 3A*). This expression pattern contrasts with that of *VRN2* in a SD-vernalization responsive variety of wheat in which *VRN2* levels remain at low SD levels after a shift to LD (*Dubcovsky et al., 2006*). In wheat, *VRN2* is down-regulated by both cold and short days suggesting that this gene might play a role in the integration of the SD vernalization and vernalization pathways (*Dubcovsky et al., 2006*). This is unlikely to be the case in *B. distachyon* as *VRN2* is upregulated during the cold (*Ream et al., 2014*) and *VRN2* down-regulation in SD is not maintained during a SD-LD shift (*Figure 4B*).

That VRN2 controls *FTL9* expression in *B. distachyon* is supported by the upregulation of *FTL9* in LD when *VRN2* expression is suppressed by amiRNAs (*Figure 4C* and *Figure 4—figure supplement 1B*) and the lack of *FTL9* expression and lack of acquisition of competence to flower in SD when *VRN2* is constitutively expressed (*Figure 4D–E* and *Figure 4—figure supplement 1C*). Thus, *VRN2*

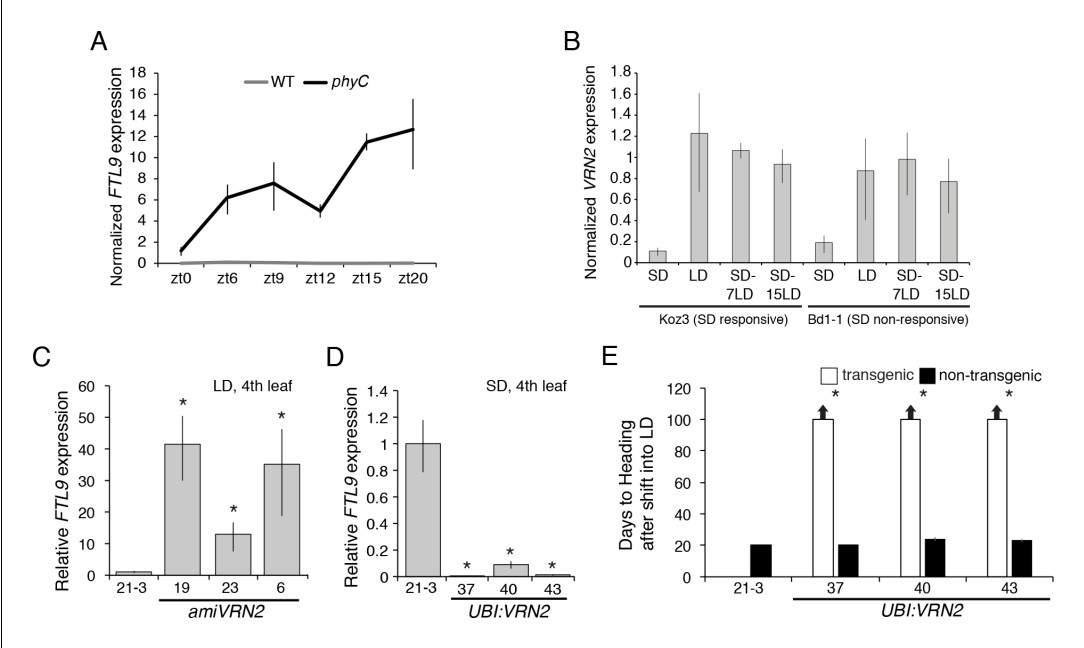

**Figure 4.** *VRN2* represses *FTL9* in LD. (**A**) Gene expression of *FTL9* in Bd21-3 (grey line) and *phyC* (black line; *phyC-1* allele was used). Plants were grown in LD (20 hr days) until the fourth leaf stage was reached at which point the newly expanded fourth leaf was harvested at zt0, zt6, zt9, zt12, zt15 and zt20. The average of three biological replicates is shown ± standard deviation (three leaves per replicate). (**B**) *VRN2* is most highly expressed in LD. *VRN2* mRNA levels in Koz3 and Bd1-1 grown solely in 8 hr SD, solely in 20 hr LD only, or 7 and 15 days after a shift from SD to LD. Samples as described in *Figure 3A*. See *Figure 3A* legend for description of the state of the meristem throughout experiment. (**C**) *FTL9* is expressed in LD when *VRN2* expression is reduced. *FTL9* mRNA levels were assessed by qRT-PCR in samples from a newly expanded leaf of Bd21-3 and *amiVRN2* plants at the fourth-leaf stage grown in 16 hr LD. (**D**) FTL9 expression is reduced in SD if *VRN2* is constitutively expressed. *FTL9* mRNA levels were determined as in C. (**C,D**) Average relative *FTL9* expression is shown for four biological replicates ± standard deviation (four leaves per replicate). Asterisk indicates a p-value < 0.05. See *Figure 4—figure supplement 1* for *VRN2* expression. (**E**) Constitutive *VRN2* expression blocks the SD vernalization response. Days to heading for three independent *amiVRN2* transgenic lines (white bars) grown in SD for 10 weeks before shifting to 14 hr LD; wt Bd21-3 and segregating non-transgenic plants (black bars) show a normal SD vernalization response. Bars represent the average of 12 plants ± standard deviation.
DOI: https://doi.org/10.7554/eLife.42153.016

The following figure supplement is available for figure 4:

**Figure supplement 1.** *FTL9* and *VRN2* expression in different photoperiods and in *amiVRN2* and *UBI:VRN2* transgenics.
DOI: https://doi.org/10.7554/eLife.42153.017

is a repressor of flowering that creates a requirement for vernalization by suppressing flowering when the days are sufficiently long, and, when the day-length decreases below a threshold in winter, the lack of *VRN2* expression permits *FTL9* expression which leads to competence to flower. The regulation is one way: over- or under-expression of *FTL9* does not affect *VRN2* mRNA levels (*Figure 2—figure supplement 3D*).

The repression of *FT* family genes by LD-upregulated *VRN2* appears to be conserved in grasses. For example, in rice, in which SD are inductive for flowering, expression of the *VRN2* ortholog *GHd7* (Woods et al., 2016) is reduced in SD enabling expression of Hd3 which encodes a florigen (*Xue et al., 2008*). Also, in wheat and barley, the florigen *FT3* is expressed in SD, and this SD-specific expression of *FT3* is correlated with lower levels of *VRN2* expression in SD (*Dubcovsky et al., 2006*; *Kikuchi et al., 2009*; *Casao et al., 2011b*). Furthermore, in barley varieties with loss of function mutations in *VRN2* results in elevated *FT3* expression under LD conditions that do not typically induce its expression (*Casao et al., 2011b*).

## Convergence between the SD vernalization and cold-mediated vernalization pathways

To explore the relationship between SD vernalization and cold-mediated vernalization, we evaluated the expression of the key flowering genes *VRN1* and *FT1* (known as *VRN3* in wheat, *Yan et al.,*

*2006*) in SD, LD, and during a SD-LD shift (*Figure 5*). *FT1* and *VRN1* are lowly expressed in SD only and LD only in Koz3 and Bd1-1 consistent with the delayed flowering phenotype of these accessions without prior SD or cold exposure (*Figure 5*). Furthermore, dissections on the parent culm revealed that the meristems were vegetative in both Koz3 and Bd1-1 when grown only in SD or in LD for the duration of the experiment. However, prolonged exposure of Koz3 (SD vernalization responsive) to SD followed by a shift to LD (SD-LD) results in the up-regulation of both *FT1* and *VRN1*, whereas this treatment does not result in *FT1* and *VRN1* expression in the SD non-responsive accession Bd1-1 (*Figure 5*). In Koz3 all of the meristems had converted to floral meristems by day 15 after shifting into LD whereas the meristems remained vegetative in the Bd1-1 accession. Thus, SD vernalization provides competence to flower by enabling the expression of *VRN1* and *FT1* once plants are in LD. As previously shown for cold-mediated vernalization (Woods et al., 2016), SD-mediated vernalization is attenuated in *amiVRN1* transgenic lines (*Figure 5C,D*), further corroborating that the cold- and SD-mediated vernalization pathways converge on enabling expression of genes like *VRN1* when plants are exposed to inductive LD. It is important to note that the loss of *FTL9* activity has no effect on cold-mediated vernalization (*Figure 2*) demonstrating that there are features that define the two pathways as separate. It will be interesting to determine mechanisms through which *FTL9* expression in SD enables an alternate path to cold-mediated vernalization to provide competence to flower.

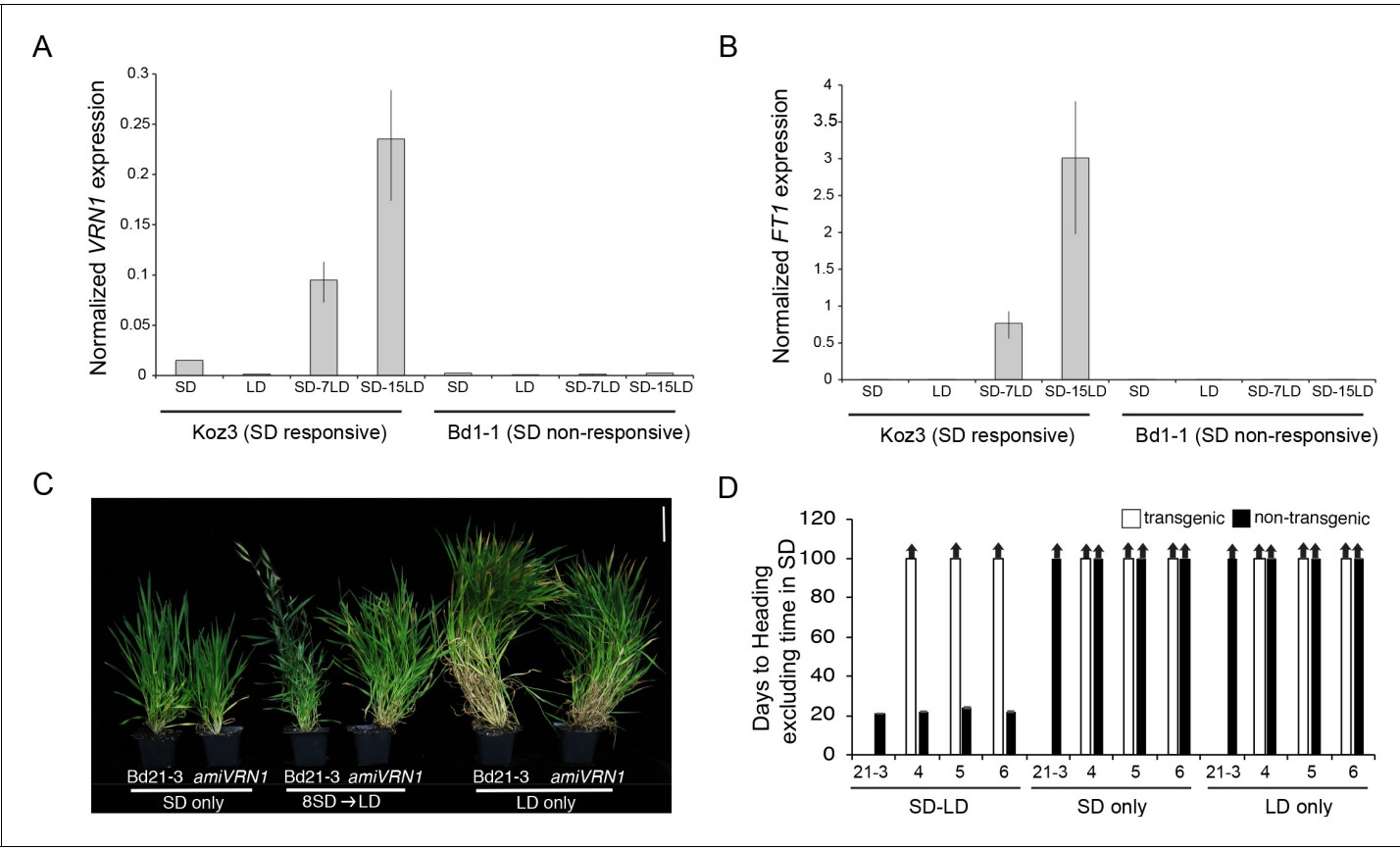

**Figure 5.** SD vernalization induces the floral promoting genes *FT1* and *VRN1* in LD and the SD vernalization response depends on *VRN1* expression. (**A**) *VRN1* and (**B**) *FT1* mRNA levels in Koz3 and Bd1-1 grown solely in 8 hr SD, solely in 20 hr LD, or 7 or 15 days after a shift from 8 weeks in SD to LD. See *Figure 3A* legend for description of the state of the meristem throughout experiment. RNA was prepared from newly expanded leaves on the parent culm. Values represent the average of three biological replicates ± standard deviation (four leaves per replicate). The experiment was repeated with similar results. Expression normalized to *UBC18* as in *Ream et al. (2014)*. (**C**) Knockdown of *VRN1* expression blocks the SD vernalization response. Representative photo of Bd21-3 wild-type, and *amiVRN1* grown in 8 hr photoperiod (SD) only, 14 hr photoperiod (LD) only and after a shift from 8 weeks in SD to LD. Bar = 5 cm. (**D**) Days to heading for three independent *amiVRN1* transgenic lines (4,5,6; white bars) grown in SD for 10 weeks before shifting to 14 hr LD; wt Bd21-3 and segregating non-transgenic plants (black bars) show a normal SD vernalization response. Bars represent the average of 12 plants ± standard deviation.

DOI: https://doi.org/10.7554/eLife.42153.018

This competence is due, at least in part, by enabling *VRN1* expression to increase in LD. For example, *FTL9* may be involved in modifying *VRN1* chromatin during SD to allow its activation during LD. A similar model applies to the cold-mediated vernalization pathway: for example, repression of *VRN1* before cold-mediated vernalization is associated with high levels of H3K27me3 at *VRN1* chromatin in both barley and *B. distachyon* (*Oliver et al., 2009*; *Lomax et al., 2018*; *Woods et al., 2017b*).

### SD-mediated vernalization in other groups of plants

Similar to cold-mediated vernalization, the SD-mediated vernalization response is found in an array of species spanning flowering plant diversification (*Chouard, 1960*; *Heide, 1994*). Advances in understanding the molecular underpinnings of cold-mediated vernalization in several different plant groups as well as knowledge from paleobotany and earth climate history indicate that cold-mediated vernalization likely evolved independently multiple times as flowering plants were radiating some 140 million years ago (e.g., *Bouché et al., 2017*). The fact that *FTL9* appears to be grass-specific but SD-mediated vernalization exists in other groups of plants suggests that SD-mediated vernalization likely evolved independently several times as well. It will be interesting to evaluate the molecular basis of SD-vernalization pathways in different plant groups.

## Materials and methods

### Growth conditions and plant phenotyping

Seeds were imbibed overnight in distilled water at 5°C then grown in MetroMix 360 (Sungrow) in 3-inch plastic pots under four 5000 K T5 fluorescent bulbs (300 mmol m–2 s–one at plant level) at 21–22°C during the light period and 18°C during the dark and fertilized weekly with Peters Excel 15-5-15 Cal-Mag and Peters 10-30-20 Blossom Booster (RJ Peters). To minimize light intensity differences, plant positions were rotated several times per week. Flowering time was measured as the number of days from the emergence of the coleoptile to the day when emergence of the spike was visible. The developmental stage of the plant was recorded as the number of primary leaves derived from the parent culm at the time of heading. For all experiments at least six plants were used to obtain the days to heading and leaf count averages. Representative phenotyping results are presented, each of which were repeated in independent experiments with similar results.

### Marker development

Development of indel markers (*Supplementary file 4*) was done using the sequenced genomes of Koz3, Bd1-1, 12 c, Bd29-1, and RON2 (*Gordon et al., 2017*). All markers are optimized to anneal at 51°C and produce a product roughly 100 bp in length. Markers were resolved using a 3% sodium borate agarose gel.

### Generation of *amiFTL9* and *UBIFTL9, UBIFTL10* Transgenic lines

amiRNAs specific for two distinct parts of the *FTL9* mRNA were designed using the amiRNA designer tool at wmd3.weigelworld.org based on MIR528 from rice in the pNW55 vector. Gateway-compatible *amiFTL9* PCR products were recombined into pDONR221 using Life Technologies BP Clonase II following the manufacturer's protocol. Clones were verified by sequencing. The pDONR221 vector containing the desired amiRNA in combination with another vector containing the maize ubiquitin promoter were both recombined into destination vector p24GWI (designed by Devin O'Connor at the Plant Gene Expression Center, Albany, CA) using Life Technologies LR clonase II plus following the manufacturer's protocol. Clones were verified by sequencing to ensure that the maize ubiquitin promoter was upstream from the amiRNA. The constructs were transformed into *A. tumefaciens* strain Agl-1. Plant callus transformation was as previously described (*Vogel and Hill, 2008*). Independent transgenic lines were genotyped for the transgene using an amiRNA forward primer specific for the targeted transcript and a reverse primer derived from the pNW55 backbone sequence (*Supplementary file 5*). Primers used to generate the amiRNAs are listed in *Supplementary file 5*.

FTL9 cDNAs were amplified from SD-grown Bd21, Koz3, and Bd1-1 plants. *FTL10* cDNAs were amplified from Bd21-3 grown in LD. cDNAs were gel extracted (Qiagen) and cloned using CloneJET

PCR according to the manufacturer's protocol (Thermo Fischer Scientific). Clones were verified by sequencing. The cDNA was then subcloned using primers compatible with the Gateway BP clonase system according to the manufacturer's protocol (Thermo Fischer Scientific). The BP plasmid was verified by sequencing and then recombined into pANIC10A (*Mann et al., 2012*) using Life Technologies LR Clonase II following the manufacturer's protocol. Clones were verified by sequencing in pANIC10A and then transformed into *Agrobacterium tumefaciens* strain Agl-1. Plant callus transformation was performed as described (*Vogel and Hill, 2008*). Independent transgenic lines were genotyped for the transgene using a cDNA-specific forward and pANIC vector AcV5 tag reverse primer (*Supplementary file 5*). Primer pairs used to clone each cDNA are listed in *Supplementary file 5*.

## RNA expression analysis and qPCR
RNA extraction and expression analysis were performed as described in *Ream et al. (2014)*.

## QTL analysis
QTL analysis was performed as described in *Woods et al. (2017a)*.

## Phylogenetic and syntenic analyses
Phylogenetic analyses of *FT*-like genes were performed using both the full-length *FTL9* and *FT* genes and the PEBP domains as seed sequences for BLAST searches using Phytozome and NCBI as described in *Woods et al. (2011)*. Maximum-likelihood analyses were conducted using SeaView 4.5.4 (*Gouy et al., 2010*). Phylogenetic analysis of the SD-vernalization interval across 51 *B. distachyon* accessions was done using variants called within a VCF file generated as in *Gordon et al. (2017)*. Relationships among *B. distachyon* accessions based on the mapped interval were determined using the TASSEL five software package (*Bradbury et al., 2007*). Synteny in the chromosomal regions around *FTL9* of *Brachypodium distachyon*, *Oryza sativa*, *Seteria italica*, *Sorghum bicolor*, *Panicum hallii*, and *Zea mays* was investigated using the GEvo software package within CoGe (*Lyons and Freeling, 2008*).

## LD only and SD only controls for characterization of the SD-vernalization response
To determine if prolonged growth in short days followed by a shift into long days (SD-LD) can promote flowering, we grew 43 delayed flowering accessions in 8 hr short days for 8 weeks before shifting into either 16 or 20-hr days (*Supplementary file 1–2*). The two controls for this experiment were accessions grown in SD and LD only. The SD only control demonstrates that growth in SD is indeed non-inductive for flowering because growth for 150 days does not permit flowering in any of the accessions tested (*Supplementary file 1–2*). Dissections on the parent culm revealed that the meristems were still vegetative after 150 days of growth in SD in all 51 accessions tested, consistent with previous reports (*Woods et al., 2014a*; *Gordon et al., 2017*). The LD only control was chosen to ensure that if robust flowering occurs following a shift from SD to LD, this was indeed due to the prior SD treatment and not simply due to the age of the plant when shifted into inductive LD. Thus, an accession is SD vernalization responsive if the SD-LD shifted plants flower much more rapidly than the LD only controls.

## Control for day and night temperature during SD vernalization
To ensure the acceleration of flowering by prolonged exposure to SD was indeed due to the shorter photoperiod and not due to extended periods of cooler dark temperatures of 18℃, we grew four accessions (KOZ3, 12 c, RON2 and Bd18-1) in SD under constant day and night temperatures of 21℃. Accessions grown under constant 21 ℃ day/night temperatures showed the same robust SD vernalization response as those grown under 21℃/18 ℃ day/night temperatures (data not shown), indicating that the SD vernalization phenomenon is due strictly to the photoperiod and not a composite of photoperiod and temperature.

## Acknowledgements

We thank John Vogel and Sean Gordon for leading the effort to sequence and analyze 51 *B distachyon* genomes (*Gordon et al., 2017*) which provided data critical for this work. We thank Jill Mahoy and Heidi Kaeppler for providing transgenic *B distachyon* lines, and Scott Woody for many helpful suggestions on improving this manuscript. This material is based upon work supported in part by the Great Lakes Bioenergy Research Center, US Department of Energy, Office of Science, Office of Biological and Environmental Research under Award Numbers DE-SC0018409 and DE-FC02-07ER6449. Funding was also provided by the National Science Foundation to RA (IOS-1258126), and a National Institutes of Health-pre-doctoral training grant to the University of Wisconsin Genetics Training program. The funders had no role in study design, data collection and interpretation, or the decision to submit the work for publication.

## Additional information

### Competing interests

Richard Amasino: Reviewing editor, *eLife*. The other authors declare that no competing interests exist.

### Funding

| Funder | Grant reference number | Author |
| --- | --- | --- |
| National Science Foundation | IOS-1258126 | Richard Amasino |

The funders had no role in study design, data collection and interpretation, or the decision to submit the work for publication.

### Author contributions

Daniel Woods, Conceptualization, Resources, Data curation, Formal analysis, Supervision, Investigation, Methodology, Writing—original draft, Writing—review and editing; Yinxin Dong, Formal analysis, Validation; Frederic Bouche, Data curation, Software, Formal analysis, Methodology, Writing—review and editing; Ryland Bednarek, Resources, Formal analysis; Mark Rowe, Conceptualization, Formal analysis, Validation; Thomas Ream, Conceptualization; Richard Amasino, Conceptualization, Resources, Supervision, Funding acquisition, Writing—review and editing

### Author ORCIDs

Daniel Woods  http://orcid.org/0000-0002-1498-5707
Richard Amasino  http://orcid.org/0000-0003-3068-5402

### Decision letter and Author response

Decision letter https://doi.org/10.7554/eLife.42153.025
Author response https://doi.org/10.7554/eLife.42153.026

## Additional files

### Supplementary files

• Supplementary file 1. Days to heading in Short days (SD,8h), long days (LD, 16 hr) and SD shifted LD plants (SD-LD). Also contains number of leaves on parent culm at time of flowering. * asterisk indicates some plants within treatment did not flower after 150 days of growth. >greater than sign indicates none of the plants flowered after 150 days of growth. SD-LD plants were first grown in 8 hr short days (SD) for 8 weeks prior to shifting into 16 hr long days (LD). Days to heading includes the time the plants were grown in SD.
DOI: https://doi.org/10.7554/eLife.42153.019

• Supplementary file 2. Days to heading in Short days (SD,8h), long days (LD, 20 hr) and SD shifted LD plants (SD-LD). Also contains number of leaves on parent culm at time of flowering. * asterisk

indicates some plants within treatment did not flower after 150 days of growth. >greater than sign indicates none of the plants flowered after 150 days of growth. SD-LD plants were first grown in 8 hr short days (SD) for 8 weeks prior to shifting into 16 hr long days (LD). Days to heading includes the time the plants were grown in SD.

DOI: https://doi.org/10.7554/eLife.42153.020

• Supplementary file 3. Days to heading in 16-hr days with and without 6 weeks or 14 weeks of vernalization. >greater than sign indicates plants did not flower after 150 days of growth.

DOI: https://doi.org/10.7554/eLife.42153.021

• Supplementary file 4. Indel markers for Koz3 X Bd1-1 F2 mapping population. Precise location differs slightly from location used in primer name due to change in the genome version used, v2.1 versus v3.1, locations will likely continue to change as new versions of the *B. distachyon* genome are released.

DOI: https://doi.org/10.7554/eLife.42153.022

• Supplementary file 5. Primers used in amiRNA, qPCR, sanger sequencing FTL9, and cloning.

DOI: https://doi.org/10.7554/eLife.42153.023

### Data availability

All data generated or analysed during this study are included in the manuscript and supporting files.

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
