## [Decision Letter]

Thank you for submitting your article "A florigen paralog is required for short-day vernalization in a pooid grass" for consideration by *eLife*. Your article has been reviewed by two peer reviewers, and the evaluation has been overseen by Hao Yu who is a member of our Board of Reviewing Editors and Christian S. Hardtke as the Senior Editor. The reviewers have opted to remain anonymous.

The reviewers have discussed the reviews with one another and the Reviewing Editor has drafted this decision to help you prepare a revised submission.

Summary:

This paper investigates the mechanism underlying the poorly understood phenomenon of short-day induced vernalization in the pooid grass *Brachypodium distachyon*. The authors study why many accessions of *Brachypodium* have the ability to perceive seasonal cues important for the release of flowering repression during winter, not only by perceiving cold signal, but also by responding to short photoperiods. The authors identify that an FT-like gene called FTL9 is expressed in short days and required for conferring *Brachypodium* accessions a rapid flowering response when they are transferred to long days. They further demonstrate that accessions without this flowering response carry a characteristic amino acid change in a conserved region of FTL9, suggesting that allelic variation of FTL9 contributes to natural variation for *Brachypodium*'s flowering response to short days. In addition, the authors show that FTL9 expression is repressed in long days by VRN2, and its effect on flowering under short days is mediated by two floral activating genes VRN1 and FT1.

Taken together, this paper presents some mechanistic understandings of short-day vernalization, which has previously been described only at the physiological level. The results also provide a new example of how different FT paralogs can evolve new functions to expand the plant ability to fine-tune their flowering time in response to different environmental conditions. The physiological and molecular genetic data are clearly presented and well designed, and the overall findings are of considerable general interest. We suggest the following revisions before this manuscript could be further considered in *eLife*.

Essential revisions:

1) We suggest that the authors perform microscopic analysis of meristems during short day treatment of responsive lines versus the lines with the inactive FTL9 allele. Currently scoring of flowering is carried out purely at the whole plant level by counting the number of days until "the emergence of the spike was visible". However, FTL9 might promote the floral transition under short days, but the spikes might not visibly emerge until the plants are transferred to long days. Such a result would alter the conclusion in this manuscript as it would indicate that flowering actually occurs in short days, which, however, is invisible because the spike does not grow out until the plant is exposed to long days. Thus, it is useful and important to dissect meristems during the short-day treatment and after the shift to long days in a short-day responsive and a short-day non-responsive variety. Such a dataset would add another level of phenotypic analysis to the paper and would strengthen the conclusion that an active FTL9 allele is required for the acquisition of floral competence without a direct effect on inducing flowering.

2) All short-day non-responsive types seem to harbor the same FTL9 allele. The text and the results shown in Figure 1—figure supplement 2 suggest that this is an inactive allele caused by a single amino acid change in a highly conserved residue. However, if there is strong selection for a straightforward loss of function allele in certain environments, as suggested, then we would expect many different alleles to arise independently in different populations, as for example, shown for FLC and FRI in populations of Brassicaceae species. This presence of the same FTL9 allele in all populations that have lost the response suggests that it might not simply be a complete null but might retain some function that causes this specific allele to be selected for whenever short-day vernalization is lost. Some discussions on the properties of the "loss of function" FTL9 allele along these lines would be worthwhile.

3) The results on FTL10 shown in Figure 3 are not convincing to indicate its florigen activity. The native FTL10 expression in *Brachypodium* is not modulated by day-length and is also very low in all conditions tested. Although constitutive expression of FTL10 induces flowering under tissue culture conditions and short days, this might not reflect its endogenous function relevant to control of flowering time. Thus, the conclusion on FTL10 function in the context of Figure 3 should be revised by either showing more pieces of evidence or toning down the claim on its role.

4) To make the paper as accessible as possible to a broad readership, it would be useful to ensure that all gene products are properly introduced on first mention. For example, in subsection “FTL9 is repressed in LD by VRN2”, *VERNALIZATION2* is first mentioned without a description of the nature of its protein product. We suggest citing the original paper(s) describing the isolation of the VRN2 gene. Introduction for some other genes is similarly lacking in this manuscript.

5) Some figure panels are not cited in order in the text. For example, Figure 4E is mentioned in the text before Figure 4A-D, while Figure 3D and E are mentioned after Figure 4. The authors must re-arrange the figure panels to describe them in an appropriate order in the text.

---

## [Author Response]

Essential revisions:1) We suggest that the authors perform microscopic analysis of meristems during short day treatment of responsive lines versus the lines with the inactive FTL9 allele. Currently scoring of flowering is carried out purely at the whole plant level by counting the number of days until "the emergence of the spike was visible". However, FTL9 might promote the floral transition under short days, but the spikes might not visibly emerge until the plants are transferred to long days. Such a result would alter the conclusion in this manuscript as it would indicate that flowering actually occurs in short days, which, however, is invisible because the spike does not grow out until the plant is exposed to long days. Thus, it is useful and important to dissect meristems during the short-day treatment and after the shift to long days in a short-day responsive and a short-day non-responsive variety. Such a dataset would add another level of phenotypic analysis to the paper and would strengthen the conclusion that an active FTL9 allele is required for the acquisition of floral competence without a direct effect on inducing flowering.

This is a great point and we completely agree. In fact, we had already done this; however, we now realize that this was not properly emphasized in the manuscript but rather was “hidden” in the Materials and methods section. We now state in the main text of the manuscript (Results and Discussion section) that “Flowering time was measured as days to heading and leaf number on the parent culm. Furthermore, dissections on the parent culm were made at the end of the experiment in accessions that did not visibly flower to determine the developmental state of the meristem.” We also now state in the main text (Results and Discussion section) “Dissections on the parent culm revealed that the meristems were all still vegetative after 150 days of growth in SD in all 51 accessions tested, consistent with our previous studies […]” We felt that it was most important to determine the state of the meristem in all accessions at the end of the SD treatment to determine if they were in fact vegetative. We also direct readers in the Figure 1 legend to see Supplementary file 1 and Supplementary file 2 which includes both days to heading as well as leaf-count data. We also now add information in Figure 3A on the state of the meristem in Koz3 (SD vernalization responsive) and Bd1-1 (SD vernalization non-responsive) during growth in SD, LD and after the shift to long days. We state in the legend “Dissection of meristems on the parent culm revealed that all meristems remained vegetative in both Koz3 (12 dissected meristems) and Bd1-1 (12 meristems) in SD and LD grown plants. After a shift from SD to LD at day 7 all of the meristems were vegetative in Koz3 and Bd1-1 (12 meristems for each accession); however, by day 15 in LD all of the meristems had converted to floral meristems in Koz3 (12 meristems) whereas they remained vegetative in Bd1-1 (12 meristems).” We refer the reader to this figure legend for details on the state of the meristem for Figure 4A and Figure 5A,B. In the main text we added (subsection “Convergence between the SD vernalization and cold-mediated vernalization pathways”) “Furthermore, dissections on the parent culm revealed that the meristems were vegetative in both Koz3 and Bd1-1 when grown only in SD or in LD for the duration of the experiment.” Also “In Koz3 all of the meristems had converted to floral meristems by day 15 after shifting into LD whereas the meristems remained vegetative in the Bd1-1 accession.”

In addition to dissecting meristems for nearly all the experiments we not only counted days to heading but also the number of leaves formed on the parent culm as a metric for flowering which we indicate in the Materials and methods section as “The developmental stage of the plants was recorded as the number of primary leaves derived from the parent culm at the time of heading.” Counting leaves also reflects the state of the meristem as once a vegetative meristem converts to a floral meristem no new leaves on the tiller will be made. In cases where we report a non-flowering phenotype there are higher leaf numbers compared to accessions that flowered at an early date indicating that the meristem in the parent culm is still vegetative.

2) All short-day non-responsive types seem to harbor the same FTL9 allele. The text and the results shown in Figure 1—figure supplement 2 suggest that this is an inactive allele caused by a single amino acid change in a highly conserved residue. However, if there is strong selection for a straightforward loss of function allele in certain environments, as suggested, then we would expect many different alleles to arise independently in different populations, as for example, shown for FLC and FRI in populations of Brassicaceae species. This presence of the same FTL9 allele in all populations that have lost the response suggests that it might not simply be a complete null but might retain some function that causes this specific allele to be selected for whenever short-day vernalization is lost. Some discussions on the properties of the "loss of function" FTL9 allele along these lines would be worthwhile.

This is a good point we now state in the subsection “Loss of the SD-vernalization response arose once in *B. distachyon*”:

“This suggests that the *FTL9* variant associated with loss of SD responsiveness arose once and was then maintained by positive selection throughout outcrossing to diverse accessions. Although, *B. distachyon* is typically inbreeding, outcrossing occurs as well (Sancho et al., 2018). That a lesion in *FTL9* that appears to have undergone positive selection arose once is in contrast to the loss of a vernalization requirement in *Arabidopsis* which arose independently many times due to loss-of-function mutations in *FRIGIDA* (Le Corre et al., 2002). Perhaps the T94K change does not result in a complete loss of function, but rather retains a function distinct from its role in SD vernalization.”

3) The results on FTL10 shown in Figure 3 are not convincing to indicate its florigen activity. The native FTL10 expression in Brachypodium is not modulated by day-length and is also very low in all conditions tested. Although constitutive expression of FTL10 induces flowering under tissue culture conditions and short days, this might not reflect its endogenous function relevant to control of flowering time. Thus, the conclusion on FTL10 function in the context of Figure 3 should be revised by either showing more pieces of evidence or toning down the claim on its role.

We are grateful to receive this comment because we do not want this key point to be misunderstood. We are NOT making any claims about the "endogenous function [of FTL10] relevant to control of flowering time." Indeed, expression data to date is consistent with no function - the RNA is barely detectable by PCR under a range of conditions - and FTL10 may in effect be a pseudogene. Nevertheless, the coding region is intact and the FTL10 protein can trigger flowering unlike FTL9 which causes competence. This is a key point of divergence of some cereal crops and *Brachypodium*: in some cereal crops the FTL10 ortholog is robustly expressed in short days and this can account for flowering in short days which does not occur in *Brachypodium*. It is important to keep in mind that not all genes with intact coding regions have roles. Evolution is a continuum and any given “snapshot” of this continuum is likely to reveal “pseudogenes” which have lesions in coding regions or regulatory elements.

We have revised the manuscript to make this point more clear. We now state in the subsection “Roles of SD expressed FT-like genes in grass flowering”:

*“FT3* is up-regulated in SD in barley and wheat, whereas *FTL10* expression in *B. distachyon* is not modulated by day-length and its expression level is quite low in all photoperiod and temperature conditions tested (Figure 3D). Due to its minimal expression, *FTL10* may not play a role in *B. distachyon* flowering. However, constitutive expression of *FTL10* from the maize ubiquitin promoter results in flowering during transgenic line generation in tissue culture (Figure 3E) and rapid flowering in SD (Figure 2—figure supplement 3I, J) similar to the effects of expression of other *FT1* orthologs in *B. distachyon* (Figure 2—figure supplement 3I, J; Ream et al., 2014) and wheat (Lv et al., 2014) indicating that *FTL10* has the ability to induce flowering. Thus, the *FT3/FTL10* orthologs in this grass-specific duplication event have diverged in barley and wheatversus *B. distachyon*. In barley and wheat, *FT3* has retained its presumably ancestral role of being expressed in SD which leads to flowering in SD. *B. distachyon* cannot flower in SD because its ortholog, *FTL10*, is no longer expressed. As discussed above, constitutive expression in *B. distachyon* of the other duplicated gene, *FTL9*, does not induce flowering in SD or in tissue culture because its role is to establish competence to flower as opposed to the florigen-like role of *FTL10/FT3* in inducing flowering (Figure 2—figure supplement 3G, I, J).”

4) To make the paper as accessible as possible to a broad readership, it would be useful to ensure that all gene products are properly introduced on first mention. For example, in subsection “FTL9 is repressed in LD by VRN2”, VERNALIZATION2 is first mentioned without a description of the nature of its protein product. We suggest citing the original paper(s) describing the isolation of the VRN2 gene. Introduction for some other genes is similarly lacking in this manuscript.

We now add some introductory information about the genes discussed upon first mention.

Subsection “Genetic mapping of the SD vernalization response identifies an FT paralog” “The founding member of this family is *FLOWERING LOCUS T (FT*) which encodes a small protein with similarity to phosphatidylethanolamine-binding proteins […]”.

Subsection “Genetic mapping of the SD vernalization response identifies an FT paralog” “[*…] VERNALIZATION1 (VRN1*), a MADS box transcription factor related to the *APETALA/FRUITFUL* family of genes in *Arabidopsis thaliana* which is a marker of the vernalized state in grasses (Woods et al., 2014).”

Subsection “FTL9 is repressed in LD by VRN2” “PHYTOCHROME C (PHYC) is a light receptor and in *phyC* mutants […]”.

Subsection “FTL9 is repressed in LD by VRN2” “*VERNALIZATION2 (VRN2*) which contains a putative zinc finger and a *CONSTANS, CONSTANS*-like and *TIMING OF CAB1* domain (Yan et al., 2004) […]”.

*5) Some figure panels are not cited in order in the text. For example, Figure 4E is mentioned in the text before Figure 4A-D, while Figure 3D and E are mentioned after Figure 4. The authors must re-arrange* the figure panels to describe them in an appropriate order in the text.

We moved Figure 4E to the Figure 4A slot and adjusted the rest of the figure so now everything is stated in order in the main text. We also move the sections “Loss of the SD-vernalization response arose once in *B. distachyon*” and “Roles of SD expressed FT-like genes in grass flowering” before the “FTL9 is repressed in LD by VRN2” section so now all of the figure panels are cited in order in the text.